

# Gene drives as a new quality in GMO releases—a comparative technology characterization

Johannes L. Frieß[1], Arnim von Gleich[2] and Bernd Giese[1]

[1] Institute for Safety/Security and Risk Sciences, University of Natural Resources and Life Sciences, Vienna (BOKU), Austria
[2] Department of Technology Design and Development, Faculty of Production Engineering, University of Bremen, Germany

## ABSTRACT

Compared to previous releases of genetically modified organisms (GMOs) which were primarily plants, gene drives represent a paradigm shift in the handling of GMOs: Current regulation of the release of GMOs assumes that for specific periods of time a certain amount of GMOs will be released in a particular region. However, now a type of genetic technology arises whose innermost principle lies in exceeding these limits—the transformation or even eradication of wild populations. The invasive character of gene drives demands a thorough analysis of their functionalities, reliability and potential impact. But such investigations are hindered by the fact that an experimental field test would hardly be reversible. Therefore, an appropriate prospective assessment is of utmost importance for an estimation of the risk potential associated with the application of gene drives. This work is meant to support the inevitable characterization of gene drives by a comparative approach of prospective technology assessment with a focus on potential sources of risk. Therein, the hazard and exposure potential as well as uncertainties with regard to the performance of synthetic gene drives are addressed. Moreover, a quantitative analysis of their invasiveness should enable a differentiated evaluation of their power to transform wild populations.

Corresponding author
Johannes L. Frieß,
johannesfriess@gmx.net

## INTRODUCTION

In recent years, innovation in genetic engineering brought forth a number of technologies to manipulate the fate of entire wild type populations. These technologies rely on a super-mendelian dissemination of genetic elements within a population of sexually reproducing animal or plant species via the germline and are identified as gene drives (GD). A number of natural mechanisms contribute to this notable capability. *Sinkins & Gould (2006)* mentioned transposable elements, meiotic drive genes, homing endonuclease genes and *Wolbachia* as naturally occurring gene drives. A theoretical concept for gene drives as a method to drive a desired gene, or a set of genes into a population was already proposed in 1960 by *Craig, Hickey & VandeHey (1960)*: "Mass release of male-producing males might

be used in control operations". In those years the spread of chromosomal translocations was already proposed as a means of population control (*Serebrovskii, 1940*; *Curtis, 1968*). *Hastings (1994)* suggested to use so called "selfish genes" for that purpose and a practical implementation was explored with the use of the P-element for germline transformation of *Drosophila melanogaster* by *Carareto et al. (1997)*. *Burt (2003)* suggested to use homing endonucleases for the design of self-replicating drives. Gene drives propagate even if they confer a fitness penalty, or in other words "Mathematically, drives are initially favoured by selection [...] if the inheritance bias of the drive exceeds its fitness penalty" (*Noble et al., 2018*, p. 201). Some secure their dissemination passively so that only offspring carrying genetic information of the drive will survive or be fertile. *Akbari et al. (2015)* called this type of mechanism "selective embryonic lethality". Others actively overcome the limitations of the Mendelian inheritance pattern by a distortion of allelic segregation i.e., fragmentation of chromosomes, for example resulting in an altered sex ratio. Active drives may also copy their genetic information between homologous chromosomes resulting in homozygous offspring. Such approaches were termed "active genetics" *Gantz & Bier (2015)*. If organisms have a comparably short generation time, as e.g., insects, then within a few months, a large share of the population could express a new gene drive-transmitted trait. In particular, very invasive gene drives may be able to impose functionalities that otherwise would not be prevalent onto entire populations.

Besides the opportunity to genetically modify entire wild populations, gene drives are discussed and partially already designed for the suppression of populations. If for example the functionality mediated by the drive consists in infertile offspring, an entire population may disappear. Currently, multiple applications are under consideration. Especially malaria- or dengue-carrying mosquitoes are potential targets. In agriculture, weeds and crop pests could be eradicated or endangered species could be immunized against pathogens using a GD. Two potential applications of gene drives even serve issues of nature conservation, namely the eradication of invasive animal or plant species (*Webber, Raghu & Edwards, 2015*) and the conservation of endangered species (*Esvelt et al., 2014*; *Ledford, 2015*; *European Commission, Scientific Advice Mechanism, 2017*). Although discussed in the 2016 NASEM report on gene drives (*National Academies of Sciences, 2016*), the idea to recover the sensitivity of pest species to pesticides or to remove transgenic resistances from feral populations have not been pursued in the scientific literature of the following years.

So far, gene drives have not yet been released, but the discussion is gaining momentum (*Hochkirch et al., 2017*; *Courtier-Orgogozo, Morizot & Boëte, 2017*; *Emerson et al., 2017*). In particular the development of new GD variants is closely linked to the upswing that genome editing methods have taken by the increasing application of CRISPR/Cas (Clustered regularly interspersed palindromic repeats and CRISPR-associated protein) gene scissors (*Gantz & Bier, 2015*; *Gantz et al., 2015*).

A variety of different GD approaches has been developed in recent years. While some are genetically engineered, others are based on naturally occurring mechanisms that bias heredity. Gene drives currently in development or already applied in caged tests encompass: Meiotic Drives (autosomal- or Y-linked X-shredder), Maternal Effect Dominant Embryonic Arrest (Medea) systems, Underdominance-based systems and Homing Endonuclease Genes

(HEG) -based systems, especially CRISPR/Cas. Others, like Killer-Rescue are currently only theoretically explored.

Until now, there is no option to retrieve a GD once released into the environment. And biosafety approaches tend to be more adapted for genetically modified plants. It is therefore important to develop approaches for risk assessment and risk management which consider the specific properties of plants and animals. Moreover, in conventional GMO-applications (mostly plants), the GMOs are released into the highly artificial habitats of agricultural fields. Furthermore, GM-plants are reaped annually and thereby retrieved. Releases are designed with the intention to avoid the spread and invasion of natural populations. In all these issues gene drives organisms represent the opposite: they are designed to spread, invade and persist for multiple generations or even indefinitely in wild habitats. The differences between conventional GMOs and GDOs were well examined by *Simon, Otto & Engelhard (2018)*. However, with regard to gene drives in plants, exposure may be enhanced or difficult to control due to the spread of pollen or a comparably large number of hybridisation partners. Furthermore inheritance and spread of a GD in plants can be complicated by selfing, dioecy and polyploidy. Although there are many ideas to restrict their spread and invasion or even to alleviate adverse effects, a complete reversal and restoration of the pre-existing state (and genotype) seems impossible to date. So-called self-limiting approaches may pose a partial exception to this as their mode of action is developed to result in a decrease of the gene drive's prevalence within a population. However, at the current state of knowledge a released GD has to be regarded as irretrievable. It is therefore necessary to further advance prospective approaches for the assessment of risk-relevant functionalities (and the associated uncertainties) of drive systems considered for release. The present work will address this issue in a comparative characterization of GD techniques.

## MATERIALS & METHODS

### Prospective technology characterization

An early assessment of new technologies with regard to potential sources of risks is important and useful as an approach to operationalize the requirements of precaution, because in the case of severe concerns in an early innovation phase mitigations, corrections and course changes to alternative development paths are more easily directed. In the first stages of an innovation process, the outlines of a technology are probably already known, but possible applications and affected systems are usually still vague. However, approaches for an early assessment of potential hazards and exposure as well as an assessment of different dimensions and forms of lacking knowledge regarding hazards and exposure already exist (*Ahrens et al., 2005*; *Steinfeldt et al., 2007*; *Owen et al., 2009*; *Giese & Von Gleich, 2015*; *Linkov et al., 2018*). The underlying hypothesis in our approach is, that the range and the forms of non-knowledge are not 'just there', but are to a large extend produced by the character of the technology. Depth of technological intervention and also the intensity of intervention are the first criteria to investigate the sources, the range and forms of lacking knowledge (ranging from uncertainties to ignorance) by scrutinizing their technological

origin. Depth of intervention as a criterion for prospective technology assessment was originally outlined as 'Eingriffstiefe' by *von Gleich (1989)*, based on thoughts from *Anders (1956)*, *Jonas (1979)* and *Jonas (1985)*. It applies to technologies based on the mathematical and experimental sciences for which the differentiation between the level of the phenomena and the level of natural laws (the logic behind the phenomena) is constitutive. Technologies that apply their technological intervention not at the level of phenomena but on structures that are able to control these phenomena, like atoms and fundamental particles or molecular structures and genes, generate a much higher power and farther reaching consequences. Thus, the depth of intervention can be identified as a source of technological power and range. The criterion 'depth of intervention' has proven to be applicable in synthetic chemistry (*Böschen, Lenoir & Scheringer, 2003*), in nanotechnology (*Rip, 2006*) and synthetic biology (*Grunwald, 2016*). Similar conceptualizations can be found in *Deutscher Ethikrat* (*German Ethics Council, 2011*) and in *Engelhard, Bölker & Budisa (2016)*. The German Ethics Council translates 'Eingriffstiefe' as "degree of manipulation" and focuses on three Aristotelian "ontological" concepts: matter including form, ontogenesis and capabilities *Deutscher Ethikrat* (*German Ethics Council, 2011*, p. 68 ff).

High power and high range of exposure due to a corresponding depth of intervention lead to a large extend of non-knowledge concerning possible effects. In order to provide additional information on the frequency and the corrigibility of the expected effects, the quantitative aspects of the use of the technology (intensity of intervention i.e., quantity and frequency of its use), its reliability in practice, the probability of failure and finally possible ways of limiting harm in case of failure have to be analysed.

The aim of a prospective technology characterization is not to identify every possible adverse effect of technologies. Instead, it should provide a basis for decision-making in the context of the precautionary principle (*United Nations, 1992*; *Commision of the European Communities, 2000*; *United Nations, 2000*; *European Environment Agency, 2002*). "The precautionary principle enables decision-makers to adopt precautionary measures when scientific evidence about an environmental or human health hazard is uncertain and the stakes are high" (*Bourguignon, 2015*). The precautionary principle legitimates precautionary action in cases when it is unwarrantable to wait until a risk is clear and proven, because a probably occurring disaster will then not be controllable. Preconditions for precautionary action are therefore: (a) lack of knowledge (from uncertainty to ignorance), (b) comprehensible reasons for concern (affecting extremely powerful and/or far reaching consequences), (c) a rudimentary cost-benefit analysis (in which e.g., medical applications with little less risky options are rated higher than applications in the food chain with plenty alternatives), (d) adequate measures (reaching from containment over substitution by less problematic alternatives to moratorium) (*Fischer, Jones & von Schomberg, 2006*). Our approach to operationalize the precautionary principle is based on technology characterization with the focus on technological interventions, functionalities and properties that generate high potentials for far reaching, by-trend-irreversible and global effects. Although a further differentiation of hazard related effects is highly dependent on the particular design of a drive, predictions on the extended range as the most remarkable quality in comparison to previous transgenes are already possible. We will therefore exclude

an assessment of the specific technological power and restrict our analysis to functionalities of gene drives which are relevant for exposure. Prospective technology characterization thus considers at least the following criteria:

### Technological range

For gene drives in general, the depth of intervention is much higher than in approaches for population control which are not based on genetic modifications. Gene drives constitute a manipulation of the very basis of organisms, their genetic characteristics, which enables a potentially high technological power and range. The technological range describes the potential spatio-temporal consequences of a GD, considering its lasting persistence in a population as well as the range with which it could spread across populations. Depending on their mode of action, gene drives theoretically have varying potentials to invade populations. Some approaches may even be self-limiting and disappear or at least only persist for a limited number of generations, whereas others are self-sustaining and—in addition to their ability to persist—may invade into non-target wild type populations (cp. *Alphey, 2014*). For this analysis the partial and transient occurrence of a drive is considered as a comparably low range, while the permanent replacement of a population with genetically altered specimens is considered a high range.

### Intensity of intervention

The intensity of intervention as mass or frequency of released organisms describes the necessary quantity of interventions to drive a desired trait into a targeted population. An approach requiring the released organisms to outnumber the wild type organisms or a series of releases would score as high intensity and if an initially lower percentage of the population is sufficient, it would correspond to a low intensity.

### Reliability of the technology

Reliability describes the probability of failure of the technology with regard to its intended use. Important reliability issues are e.g., linkage-loss of the cargo gene and its driver system, the generation of resistances in the target population, coevolution of a pathogen and system decay (*Alphey, 2014*).

### Corrigibility or limitation of damage in case of failure

This criterion addresses an important aspect of risk management. Can the damage of a failed GD be reversed by any means and if so, how laborious are they, compared to the initially released construct or system? For some GD-technologies it is claimed that they can be somewhat remedied by a release of wild type organisms. But such an endeavour would not really reverse the damage done. Even more difficult to estimate are corrective actions such as a reversal drive, which on one hand relies on the release of a second generation GD to remedy the failures of the first. But on the other hand, the gene pool of the target population in any case retains transgenic elements.

The criteria of prospective technology characterization including corresponding GD-specific parameters and examples are shown in Fig. 1.
| 1. Depth of Intervention | |
|---|---|
| **Technological Power (Hazard)** (Not investigated here) Genetic Modification, Invasiveness Biased Inheritance, Toxins | **Technological Range (Exposure)** Non-Mendelian Inheritance, altered Fitness, Self-Limiting/Self-Sustaining |

| 2. Intensity of Intervention | |
|---|---|
| **Mass** Size of necessary Releases | **Frequency** Number of necessary Releases |

| 3. Reliability |
|---|
| On-, Off-, Non-Target Effects, Resistance-Formation, Mutation, Linkage-Loss of Cargo and Driver, Coevolution of Pathogen |

| 4. Corrigibility |
|---|
| Proposed Options: Second Generation Drive, Overwriting Drive |

**Figure 1 Criteria of prospective technology characterization with corresponding gene drive-specific effects and options.** Technological power is not in the focus of this rather general prospective study due to the very early innovation phase, where the particular design (esp. their cargo) and application context of gene drives is not yet clear.

## Deterministic recurrence-based model for gene drive inheritance

The model was based on the inheritance schemes of the various GD techniques. The probability of the occurrence of a certain genotype was multiplied by its respective fitness. Random mating based on the respective genotypes' percentages within the infinite population was assumed. Genotype fitness and initial population percentage can be chosen by the user. To calculate the invasiveness of the GD techniques, the variation of the fitness and population percentage parameters were automated for a given generation and depicted in a colour-coded graphic where population percentages are assigned different colours, depending on chosen thresholds. See supporting information for further details.

## RESULTS

### Comparative prospective technology characterization of gene drives

In the following, meiotic drives, Killer-Rescue-systems, the Medea-approach, Underdominance as well as Homing Endonuclease-based gene drives are introduced and discussed according to the criteria of prospective technology characterization.

As a means to compare the different GD-technologies, a rough assessment according to the criteria mentioned above was applied. Regardless of the fact that such a classification has to be further differentiated in subsequent studies, it has to be noted that this rating only refers to the comparative approach between the technologies included in this study and does not reflect any conclusion on their absolute impact.

### X-Shredder as an example for meiotic drives

Meiotic Drives (MD) consist of selfish genetic elements which cause a distortion of allelic segregation that results in a bias of the frequency of Mendelian inheritance. For instance, the Mendelian segregation frequency of 50% is distorted up to 70% in *Zea mays* (abnormal chromosome 10 [Ab10]), which affects Gonotaxis (*Lindholm et al., 2016*; *Australian Academy of Science, 2017*). Other MDs have been reported for *Drosophila melanogaster* (segregation distorter [SD] system) (*Larracuente & Presgraves, 2012*), the mouse *Mus musculus* (t-haplotypes, causing a transmission ratio distortion) (*Silver, 1993*), distorted sex ratios in Silene species (*Taylor, 1994*) and mosquitoes. In the latter, MDs are naturally occurring in *Aedes aegypti* (*Craig, Hickey & VandeHey, 1960*) and *Culex pipiens* (*Sweeny & Barr, 1978*). A major drawback of MDs consists in the fact that the fitness of other alleles at the same locus, which do not bias transmission, and alleles linked to them, is reduced (*Lindholm et al., 2016*).

For gene drives, a particularly interesting MD is the so called X-Shredder, which causes fragmentation of the X chromosome by nucleases during male meiosis. Thereby only Y-bearing sperm can produce viable offspring, which is of course male (*Newton, Wood & Southern, 1976*). An autosomal X-shredder can be regarded as self-limiting, a Y-linked X-shredder as self-sustaining (*Burt, 2003*; *Burt & Trivers, 2006*; *Deredec, Burt & Godfray, 2008*). A Y-linked X-shredder can invade adjacent populations or species with incomplete mating barriers, therefore widespread effects may be anticipated (*Alphey, 2014*). *Galizi et al. (2014)* published a synthetically engineered X-shredder aiming at spermatocyte meiosis in *Anopheles gambiae*, producing mainly Y-chromosome-carrying sperm, causing a male bias of up to 95%. A distortion of the sex ratio is a penalty to fitness, which may in extreme cases lead to a population's extinction. Although rarely, in Drosophila, sometimes 100% female progeny is achieved. Therefore, this trait is highly selected against. Hence, meiotic drive-based extinction has never been observed in natural populations (*Helleu, Gérard & Montchamp-Moreu, 2015*). For this study we focus on the self-sustaining variant of the Y-linked X-Shredder.

### Range of X-Shredder intervention

The X-Shredder approach potentially constitutes a self-sustaining GD. A male bias of up to 95% would cause a major population suppression. Moreover, a population consisting mainly of males is much more likely to migrate in search of females. Taken together, there is some evidence that the technological range of X-Shredder drives can be regarded as high.

### Intensity of X-Shredder intervention

To obtain a rapid effect, the X-Shredder approach requires a mass release of males. The necessary intensity of the technological input can thus be regarded as high. However, over multiple generations even a small release size could theoretically suffice to replace a population, dependent on the fitness of the gene drive organism (GDO).

### Reliability of the X-Shredder technology

Based on the small number of available publications on X-Shredder approaches in a preliminary comparative assessment of gene drives, key points of error encompass:

- Lowered fitness of laboratory-reared GM insects due to inbreeding (colony effect).
- Selection against the fitness burden.
- Errors in the release: wild types, and phenotypic male wild types carrying the non-functioning construct (would reduce the suppressive effect).

### Possibilities for damage limitation caused by X-Shredder technology

There is no possibility to directly remedy the damages obtained from an X-Shredder release. This makes the technique highly problematic, it is built to first invade and replace, followed by immediate suppression, due to the lack of females. Its low threshold further exacerbates the handling of Y-linked X-Shredder gene drives.

## Killer-Rescue

The Killer-Rescue system was first proposed by *Gould et al. (2008)*, it consists of two unlinked loci one encoding a toxin (killer allele), the other encodes an antidote (rescue allele). Thereby, the toxin and antidote could consist of miRNAs and a recoded gene or a toxic protein and toxin-inhibiting enzyme. Furthermore, a cargo gene can be fused to the antidote gene. Homozygous carriers for both genes would be mass-released into wild populations, offspring inheriting the killer allele but not the rescue allele would be non-viable. Since both alleles are not linked in their inheritance, the killer allele will be quickly selected from the population, while the rescue allele confers a clear fitness gain and will increase in its prevalence. As soon as the killer allele completely disappeared from the population, so will the rescue allele's fitness gain. As a consequence, the rescue allele will again decline in its prevalence, unless the cargo gene confers a gain in fitness. The Killer-Rescue system is highly dependent on the fitness of the rescue and cargo genes which determines the time until the cargo genes are eliminated from a population (*Marshall & Akbari, 2017*).

### Range of Killer-Rescue intervention

The Killer-Rescue system is not by design a suppression drive. Considering the technological range, Killer-Rescue, due to its non-persistent quality, its therefore limited probability of contamination of other populations and relatively high invasive threshold scores as low.

### Intensity of Killer-Rescue intervention

The Killer-Rescue system potentially reduces the population size only in dependence on the number of released GDOs. According to model scenarios by *Gould et al. (2008)*, this drive-system is reliant on a high number of released carriers of up to a ratio of GDOs to wild types of 2:1. Although this ratio is much lower than reported for other mass release-dependent techniques, the wild type population most likely has to be outnumbered to be successful. Therefore, mass and frequency of Killer-Rescue are regarded as high.

### Reliability of the Killer-Rescue technology

For Killer-Rescue based drives, reliability may be impaired by:

- Lowered fitness of laboratory-reared GM insects due to inbreeding (colony effect).
- The selection against the fitness burden (resistance formation or toxin-inactivation).

- Linkage loss between rescue and cargo gene.
- Natural evolution of an antidote or inactivation of the killer allele.

### Possibilities for damage limitation caused by Killer-Rescue technology

Since it is expected that the Killer-Rescue system has a high invasion threshold (although lower than that of other threshold-dependent techniques) the most feasible option to limit the spread of this GD is a release of wild types (*Gould et al., 2008*). Additionally, it would be recommendable to use miRNA as a killer allele in order not to give the carrier-organisms a toxic functionality.

This system is designed to be a self-limiting modification GD in which, if the cargo gene bears a fitness penalty, its prevalence in the population would decrease after a number of generations. There is a possible variant where multiple copies of the killer allele are incorporated into the GDOs' genome, enhancing the selective benefit of the rescue allele. A particular benefit of the technique is that it is easy to design and engineer (*Gould et al., 2008*).

## Medea (maternal-effect dominant embryonic arrest)

The term Medea is an acronym pointing at the sorceress in Greek mythology who killed her own children. This indication is accurate, as a Medea selfish genetic element consists of two chromosomally-located tightly linked transgenes: one that encodes a (miRNA-) toxin deposited in all embryos of Medea-bearing mothers, and a second that encodes an antidote (a silencing resistant recoded gene without the miRNA-sequence) active in the zygote (*Akbari et al., 2014*). Therefore, only Medea-bearing offspring (hetero- or homozygous) survives. This maternally induced lethality of wild type offspring not inheriting a Medea allele grants an ability to invade populations.

The Medea elements were first discovered in Tribolium flour beetles (*Beeman, Friesen & Denell, 1992*) and have also been reported in mice (*Peters & Barker, 1993*; *Weichenhan et al., 1996*). The only published synthetic Medea constructs (Medea *myd88*, *o-fut1* and *dah*) have been inserted on an autosomal chromosome in *Drosophila melanogaster*. The myd88 construct has also been introduced into the genome of the cherry fruit fly, *D. suzukii* (*Buchman et al., 2018*). *myd88* is a maternally expressed gene required for embryonic dorso-ventral pattern formation. If Medea is located on the X chromosome in a X/Y male heterogametic species, Medea is predicted to spread to allele fixation, with wild type alleles being completely eliminated (*Akbari et al., 2014*).

Medea organisms exhibit a high-frequency stable equilibrium when the transgenic construct is associated without any fitness cost (*Gokhale, Reeves & Reed, 2014*). The fitness costs of homozygote Medea Drosophila were estimated to be 27.3% and 17.4%, respectively, for two different targeted genes. In lab trials, where 25% of the original members were homozygous for Medea, the gene spread through the entire population within 10–12 generations. Observations indicate that a single copy of each Medea toxin is sufficient to induce 100% maternal-effect lethality and a single copy of each rescue transgene is sufficient to rescue normal development of embryos derived from mothers expressing one or two copies of the toxin (*Akbari et al., 2014*).

Until now, attempts to establish a Medea system for *Aedes aegypti* were not successful. Currently, Medea is planned to be applied for population control of the cherry fruit fly (spotted-wing fruit fly *Drosophila suzukii*) in California (*Regalado, 2017*). Two considered approaches are to either target female fertility genes or to alter the ovipositor of the flies to make them unable to puncture the ripening cherries. *Buchman et al. (2018)* found pre-existing native resistances against the miRNA toxins of their construct in five out of eight examined *D. suzukii* strains. Together with the high fitness penalties conferred by the construct the Medea GD now has to be considered a high threshold drive, that when conferring a large fitness penalty will only be transiently maintained in the population without supplemental releases. In a mathematical model for the *myd88* construct in the cherry fruit fly, a fitness cost for heterozygotes of 28% and 65% for homozygotes were assumed (*Buchman et al., 2018*).

### Range of Medea intervention

Dependent on the cargo gene and fitness penalty, Medea drives will either drive to fixation or will be selected from the population after a number of generations and have therefore only a potentially transient effect on population size. However, due to its potentially high invasiveness and the ability to spread into non-target populations its range and thus its potential of exposure should be regarded as high.

### Intensity of Medea intervention

It would not require many genetically-altered organisms to drive a trait to fixation if the fitness penalty is low. However, as demonstrated by *Buchman et al. (2018)*, due to the pre-existing resistances and high fitness penalties it is more likely that multiple mass releases are required for a successful drive (*Marshall et al., 2017*). Therefore, a high intensity of intervention is considered for Medea gene drives.

### Reliability of the Medea technology

Key points of error encompass:

- Lowered fitness of laboratory-reared GM insects due to inbreeding (colony effect).
- The selection against the fitness burden (resistance formation by toxin-inactivation or selection for pre-existing resistances).

### Possibilities for damage limitation caused by Medea technology

A potential measure would be to release a second generation Medea GD. This would introduce a new toxin-antidote combination as well as the antidote for the first generation toxin. Although the suppressive effect of Medea may be stopped by this approach, it introduces even more persisting GMOs into the ecosystem.

## Underdominance (UD)

Underdominance, also known as heterozygote inferiority is a genetically engineered GD technique. So far, there are two different approaches called UD[mel] (*Akbari et al., 2013*) and *Rpl14* (*Reeves et al., 2014*). In the two-Locus UD[mel]-approach, one or two maternal toxins which target maternal genes essential for embryonic development (oogenesis and

embryogenesis) are introduced within two constructs. Each construct consists of a maternal toxin gene and an embryonic antidote. However, the antidote to each toxin is located on the other construct. One construct encompasses the genes for toxin A and antidote B, the second construct holds toxin B and antidote A. An organism must receive both antidotes to the maternally administered toxins to be viable. Therefore, UD heterozygotes have a lower fitness than homozygotes (*Reeves et al., 2014*). When a UD-female heterozygous for both constructs mates with a wild type male, 25% will be heterozygous for both constructs, while 25% of offspring will be non-viable wild types, and 50% will be non-viable due to the lack of one of the necessary antidotes. With myd88, *dah* and *o-fut-1* (*Akbari et al., 2013*) the toxins of Underdominance constructs may be the same as utilized in the Medea technology. Since the toxins are administered maternally, a release of wild type males into a replaced Underdominance population would lead to a population crash, as all offspring would inherit the wrong antidote (*Akbari et al., 2013*).

The *RpL14*-approach, utilizes a haploinsufficient cytoplasmic ribosomal protein. One allele of *Rpl14* is replaced by a toxin-antidote combination of the same gene. The construct consists of a miRNA-gene against the targeted gene and a recoded variant of the gene, immune against the miRNA. An Underdominance GD requires a high threshold release (*National Academies of Sciences, 2016*). For the *RpL14* construct, this threshold is estimated to be as high as 61% of the total population (*Reeves et al., 2014*). Therefore, an unintended underdominant population transformation can theoretically be mitigated where it is realistically possible to release sufficient wild type individuals to traverse the unstable equilibrium in the lower frequency direction (*Gokhale, Reeves & Reed, 2014*).

### Range of Underdominance intervention
In comparison to the Medea approach, the range of Underdominance is estimated to be low, due to its higher invasion threshold (*Alphey, 2014*).

### Intensity of Underdominance intervention
Since an application of this technology is even more frequency-dependent than a Medea approach, requiring even greater mass releases, its intensity is also rated as high.

### Reliability of the Underdominance technology
The intended functionality of Underdominance-drives may be impaired by

- Lowered fitness of laboratory-reared GM insects due to inbreeding (colony effect).
- The selection against the fitness burden.

### Possibilities for damage limitation caused by Underdominance technology
Should the release of underdominant specimen eventually lead to an undesired population replacement (or other undesired side effects), a wild type release could theoretically shift the population percentage beneath the necessary threshold and thereby mitigate the GD.

## Homing Endonuclease Genes (HEG)
HEGs are selfish genetic elements. But different from transposable elements, they code for a restriction enzyme with a target sequence of 20–30 bp. The HEG is nestled within its

own recognition site. An expressed homing endonuclease-protein finds intact recognition sites and cuts them. Then the selfish genetic element relies on the DNA-repair mechanism of homologous recombination which copies the HEGs code and inserts it into the cut-site on the homologous chromosome. In the following we will focus on the use of the CRISPR/Cas9-system for the construction of HEG-based drives.

## CRISPR/Cas9

CRISPR stands for Clustered Regularly Interspaced Short Palindromic Repeats, while Cas stands for CRISPR-associated protein. Both components originate from an adaptive immune system of bacteria and archaea. Cas9 is a ribonucleoprotein (RNP), able to bind guide RNAs (gRNA), aka crRNA that specifically recognizes and binds to the target sequences (20 nucleotides) (*Doudna & Charpentier, 2014*). The target DNA-sequence must contain a protospacer adjacent motif (PAM) with the sequence NGG for the Cas protein to cut (*Jinek et al., 2012*; *Doudna & Charpentier, 2014*). The cut takes place three nucleotides upstream of the PAM, causing a blunt end double strand break (DSB). This technology can be used to cause deletions as well as insertions, relying on homologous recombination (*Doudna & Charpentier, 2014*). The guide RNAs serve target site recognition which makes this technology cheaper and easier to customize, while also being more effective (*Jinek et al., 2012*; *Doudna & Charpentier, 2014*) than conventional genome editing techniques and are thus also suitable as a GD. Figure 2 illustrates the mode of action of such a GD. The most probable application would utilize a CRISPR/Cas9-mediated GD system inheriting a cargo gene to the vast majority of its offspring, which would burden the population's fitness by e.g., targeting the organisms' fertility. However, low rates of homology directed repair (HDR) may hinder the spread of the drive (*Unckless, Clark & Messer, 2016*).

A non-homologous repair of the cut site by Non-Homologous End Joining (NHEJ) or Microhomology-Mediated End Joining (MMEJ) reduces the conversion rate because these alternative mechanisms often cause mutations or deletions at the target site of the endonuclease. Usually when a cut is not repaired by HDR, the result is a drive resistant allele. Depending on the genomic location, HDR vs. NHEJ efficiency could be as low as ~10% (*Lin & Potter, 2016*). To reduce these events, CRISPR/Cas9 could be used to enhance HDR gene expression and repress NHEJ-genes. This could be achieved by the inclusion of HDR-genes and NHEJ-repressor genes. Furthermore, the generation of nucleases creating sticky-end overhangs as opposed to blunt ends may optimize the repair in the target organism. The rate of HDR depends on the species, cell type, developmental stage, and cell cycle phase. For example faithful copying was achieved with up to 97% efficiency in mosquitoes but only 2% in fruit flies (*Esvelt et al., 2014*).

Other studies yielded average homing rates of 56% (*KaramiNejadRanjibar et al., 2018*) and 97% (*Gantz & Bier, 2015*) in Drosophila and 98.8% in *Anopheles stephensi* (*Gantz et al., 2015*) and even 99% in wild yeast (*DiCarlo et al., 2015a*).

Homing resistant alleles may as well occur due to random mutations, affecting the recognition site of the gRNA or the PAM. A general option to prolong the activity of a CRISPR/Cas-based drive in cases of target site mutagenesis is therefore to engineer multiple attack loci for the CRISPR/Cas9-system in the genome (multiplexing). Each

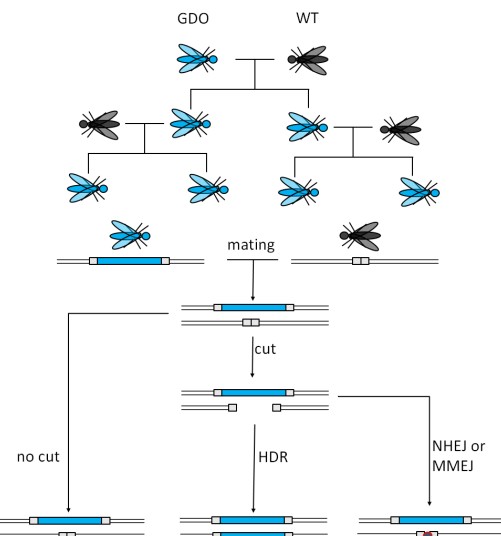

**Figure 2** **Mechanism of CRISPR/Cas9-based gene drives.** A gene drive organism carries the gene drive cassette and mates with a wild type. The gene drive cassette expresses the CRISPR/Cas9 complex, which then cuts its recognition site defined by the gRNA on the homologous chromosome. This cut can then either be repaired by homology directed repair (HDR) copying the gene drive cassette into the cut region or by mechanisms creating a homing resistant allele, like Non-Homologous End Joining (NHEJ) or Microhomology-Mediated End Joining (MMEJ) (adapted from *Esvelt et al., 2014*).

additional site reduces the chance of mutation in all attacked alleles. However, very large populations—such as those of some insects—might require unfeasibly large numbers of gRNAs to prevent resistance.

First observations suggest that resistance can also be the result of a so called "dominant maternal effect" (*Ping, 2017*). It is assumed that Cas9 deposits in the oocyte cause early cuts in the genome of the sperm cell, shortly after fusion. Upon fertilization, if sufficient Cas9 (and gRNA) is in the cytoplasm of the zygote which is homo- or heterozygous for the GD the CRISPR/Cas complex finds and cuts its recognition sites in the sperm's genome before the homologous female genome is close enough to be recruited for homologous recombination. Without a homologous template the cuts are then repaired by NHEJ and thus a resistant allele may arise. In such an event, the number of gRNA variants is meaningless. Propagation of resistant individuals may be minimized by targeting essential genes (*Noble et al., 2017*).

### *Range of CRISPR/Cas9-gene drive intervention*
In comparison to other GD techniques, CRISPR/Cas9-based GD systems are unique due to their potential to generate homozygous offspring with wild types. Since these drives seem to be self-sustaining for multiple generations despite a fitness burden their range can be regarded as high.

### Intensity of CRISPR/Cas9-gene drive intervention

Due to its non-mendelian, copy-based inheritance pattern, CRISPR/Cas9-gene drives are nearly frequency-independent. Therefore, this technology's necessary frequency is low in comparison to other techniques.

### Reliability of the CRISPR/Cas9-gene drive technology

Failure of the CRISPR/Cas9-gene drive technology may be a result of:

- Lowered fitness of laboratory-reared GM insects due to inbreeding (colony effect).
- The selection against the fitness burden.
- NHEJ and MMEJ instead of HDR.
- Incomplete or imperfect copying during HDR (if the deletion preserves the reading frame, it leads to a homing-resistant allele (*Marshall et al., 2017*)).
- Off-target effects (unspecific binding of gRNA causes unintended insertions at different loci).
- On-target misinsertions (unwanted genes or gene fragments are inserted into the target locus, instead of or additional to the desired genes) sometimes gRNA sequences are inserted (*Li et al., 2016*).
- Emergence of homing resistant alleles due to random target site mutagenesis.
- Sequence polymorphisms (resistance due to genetic variations within a species. To overcome this problem multiple gRNA variants can be added to the CRISPR/Cas cassette).
- Intragenomic interactions (the distance of gRNA target sites may affect homing rates (*Marshall et al., 2017*)).
- Maternal effects (*Ping, 2017*).
- Release of phenotypic wild types carrying the non-functioning construct (would reduce the suppressive effect and could constitute a persistently GD-resistant sub-population).

### Possibilities for damage limitation caused by CRISPR/Cas9-gene drive technology

As a potential approach to reverse detrimental damages caused by CRISPR/Cas9-gene drives the (mass) release of a secondary drive was suggested, a rescue drive which cuts out the cargo gene and forms a resistant locus (*Esvelt et al., 2014*). This however, would also mean a population replacement, persistently introducing additional synthetic genetic material into the ecosphere as the CRSIPR/Cas9-system would remain in the populations' gene pool. Some proposed drives to limit the spread are:

- Reversal (or overwriting) drive (*Esvelt et al., 2014*).
- Immunizing drive (pre-emptively) to make populations immune to another drive (*Esvelt et al., 2014*).
- Split drive to serve local confinement (*DiCarlo et al., 2015b*).
- Daisy chain drive for confinement in space and time (*Noble et al., 2016*).

## Invasiveness of gene drives

The goal of this assessment is to describe the capacity of gene drives to achieve a population replacement. To this end a simple population-genetic model was chosen that yields genotype percentages as a function of fitness penalty and relative release size of GDOs (cf. *Gould et al., 2008*; *Ward et al., 2011*; *Dhole et al., 2018*). The chosen model enables a comparative analysis of different GD systems where more specific models prohibit their investigation in a common analytical frame due to the amount of necessary data and individual adaptions to the respective types of GD.

Many modelling approaches focus on CRISPR/Cas-mediated gene drives. Their focus may lie on homing or conversion rates (*Unckless et al., 2015*; *Eckhoff et al., 2017*), some models also emphasise spatial spread (*Tanaka, Stone & Nelson, 2017*) and resistance formation (*Drury et al., 2017*). The second most modelled GD system is X-shredder, focussing again on spatial spread (*Beaghton, Beaghton & Burt, 2016*), shredding rate (*Eckhoff et al., 2017*), drive efficiency in dependence of seasonal rainfall patterns (*Lambert et al., 2018*) or the tendency of suppression drives to select for inbreeding (*Bull, 2016*). Other studies on Underdominance and Daisy chain (*Dhole et al., 2018*) or Killer-Rescue drive systems (*Edgington & Alphey, 2018*) focus on migration and invasion. *Ward et al. (2011)* explore possibilities of auto- and gonosomal Medea drive systems. Lastly, combinations of GD systems were analysed by *Huang et al. (2007)* and *Gokhale, Reeves & Reed (2014)*.

As a basis, our underlying recurrence-based, deterministic model utilizes inheritance schemes in a hypothetical population of infinite size. Therefore, it is impossible to achieve a population suppression or eradication in this model. All genotypic subpopulations are regarded as relative percentages of the whole population (for a more detailed explanation see Supplemental Information).

Two-locus Underdominance, Medea, a CRISPR/Cas-mediated GD including resistance allele formation, Killer-Rescue and a Y-linked X-Shredder were chosen for a quantitative comparison of their invasiveness. As positive and negative controls, the spread of two different transgenes lacking the GD-specific functionality of super-Mendelian inheritance were calculated: (a) female specific release of insects carrying a dominant lethal (fsRIDL, *Fu et al., 2010*) and (b) a fitness gain conferring transgene e.g., a pesticide resistance, respectively. For the calculations, the following assumptions had to be made that should be taken into account for a critical discussion of the presented results on the invasiveness:

- fsRIDL (negative control): Female lethality is 100% regardless of zygosity. Cumulative fitness penalty for each allele was assumed.
- Transgene with fitness loss/gain (negative/positive control): Cumulative fitness loss/gain for each allele was assumed.
- X-Shredder: A Y-linked X-Shredder system with a male biased sex ratio of 95% was assumed according to *Galizi et al. (2014)*. Since the ratio of females cannot decrease below 5%, due to the assumptions of our model, the thresholds were adapted for this approach to 7% and 93% in the cross section computation.
- Killer-Rescue: Cumulative fitness penalties were assumed per allele regardless of killer or rescue.

 

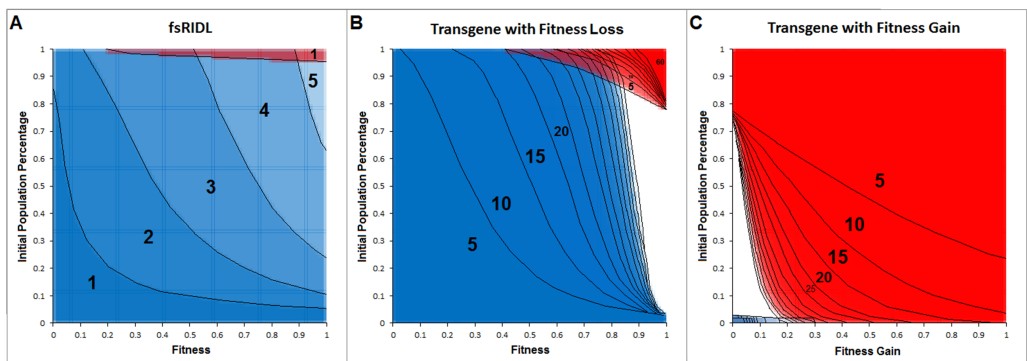

**Figure 3** Cross section overlays of the negative controls for the model represented on the fsRIDL technique and the fate of a transgene either conferring a fitness loss or gain. Red, Wild type population percentage below 5%; Blue, wild type population percentage above 95%. Black numbers and lines represent the respective generation post-release. Lines were inserted by hand for clarity.

- Medea: Fitness penalty was assumed being cumulative for hetero- and homozygous Medea-carriers.
- Underdominance: A two-locus autosome Underdominance system similar to UD$^{mel}$ (*Akbari et al., 2013*) was modelled. Female-carriers kill offspring that do not inherit at least one copy of each construct. It was assumed that heterozygosity for each of the Underdominance alleles confers a 15% fitness penalty. Therefore, the double hetero UD genotype's fitness is 30% lower than that of the double homo genotype. Whereas homozygosity in one construct but lack of the other construct results in half the fitness penalty of the double homozygotes.
- CRISPR/Cas-mediated gene drive: The homing rate was assumed to be 98% similar to data presented by *DiCarlo et al. (2015a)*. Resistance formation rates were assumed as the direct reciprocals of the homing rates, i.e., 2%. Fitness penalties were assumed to be half for heterozygous GDOs. Each resistance allele was assumed to confer a 10% fitness penalty. Homozygously resistant population percentages above 95% are depicted in green in the overlays.

Figure 3 shows positive and negative control approaches for the model as transparent overlays of cross sections for up to 60 generations post release, in 5-generational steps. Negative controls are represented by the complete fading of fsRIDL-carriers from a population within five generations and the spread of a transgene which confers a fitness loss. The spread of a transgene which confers a fitness gain represents a positive control. The blue areas represent combinations of fitness and population percentage at a given generation post-release at which more than 95% of the population is of wild type genotype. Red areas represent fitness and population percentage combinations at given generations resulting in less than 5% wild type genotypes in the population.

Figure 4 shows the allele frequency of gene drives. The depicted generations post-release range from 5 to a maximum of 60, in 5-generational steps.

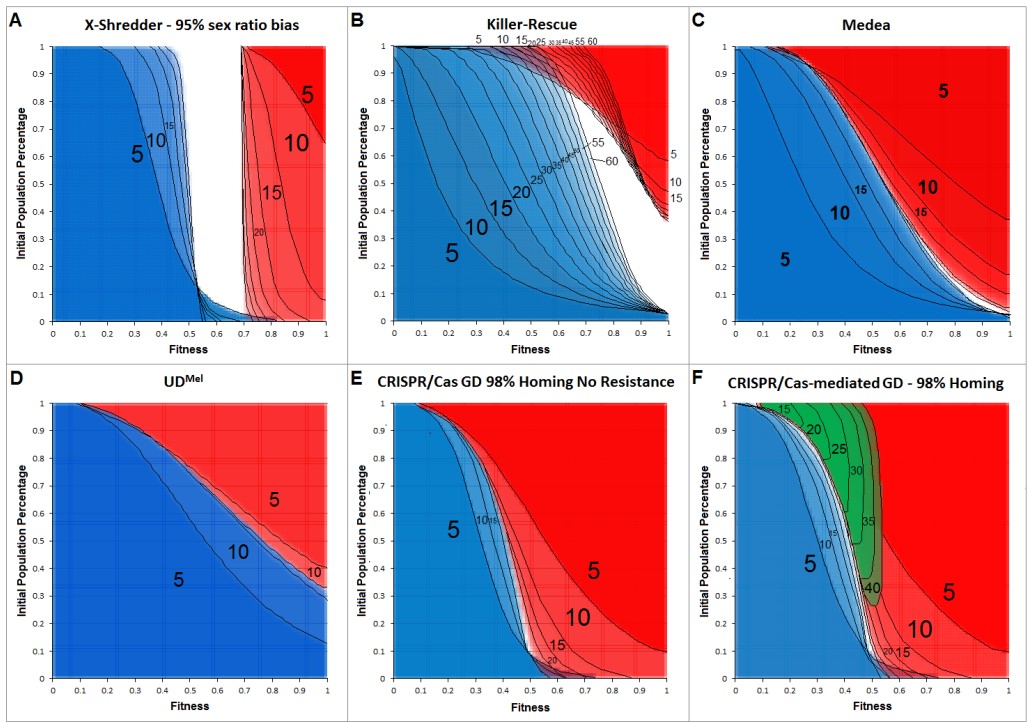

**Figure 4 Cross section overlays of fitness and initial release population percentage for different gene drive techniques.** Red, wild type population percentage below 5%; Blue, wild type population percentage above 95%. Green, population homozygous for resistance alleles above 95%. Black numbers and lines represent the respective generation post-release. Black lines were inserted by hand for clarity.

The cross-sections for invasiveness of the gene drives represent spread dynamics with clear separation from negative controls as well as the positive control (cf. Fig. 3). Calculations for a Y-linked X-Shredder show a result which is markedly different from the other gene drives investigated by this approach. Since it is linked to the Y-chromosome, X-Shredder females do not occur and wild type females are only indirectly affected by it. But all males from a mating with an X-Shredder male are also X-Shredder males. An X-Shredder drive reaches an extremely low invasion threshold. Though it takes a considerable amount of generations and the X-Shredder males' fitness must not be below 70% if an invasion threshold below 10% should be sufficient for a population invasion above 95%. Interestingly, there seems to be a persistent intermediate state for X-Shredder organisms with a fitness between 55% and 70% (white area in Fig. 4A). This intermediate state area even persists long beyond the shown 40 generations (tested for up to 150 generation post release, see Fig. S11). It is also important to note that opposed to the other gene drive techniques examined here, the X-Shredder drive is mainly applicable for suppression drives. However, as the model assumes an infinite population size a population eradication cannot be modelled.

The Killer-Rescue cross sections show that even with no fitness penalty at least a release threshold of 40% is required to achieve a population replacement. At high release

percentages the red areas are receding over the generations and in part even identify as blue areas in later generations, this underscores the transient character of the Killer-Rescue system.

For Medea obviously, the area of wild type individuals (blue) is growing faster than the area for a population with less than 5% wild type individuals (red). From this simple modelling approach, it seems as if Medea could be a candidate for a temporally limited GD application which gets lost after an active phase for e.g., 25 generations. According to the cross section overlay shown in Fig. 4 this can be achieved with lower invasion thresholds and for a longer time than with an Underdominance-based drive system.

The two-locus Underdominance approach will end up in either a population replacement or the loss of the GD construct more quickly than Medea. Also, UD is more threshold-dependent. Even with the highest fitness for the Underdominance organisms, a release threshold of approx. 40% is required.

For a CRISPR/Cas-mediated GD, a homing rate of 98% was assumed. No other gene drive examined in this work occupies a comparably large red area for comparably low invasion thresholds and the given generations post release. This definitely makes CRISPR/Cas the most invasive technique in this model. Also noteworthy is the area in the low population percentage around 55%–73% fitness which is considered a blue area after 5 generations but becomes occupied by the red area in later generations. This also indicates the high invasiveness of the technique due to its threshold-independence, as even with a release ratio below 5% of the population and a fitness down to 60% and 70%, a population replacement can be achieved. For the CRISPR/Cas system, resistance formation is an important factor that reduces its transforming capacity. In our model, the inverse of the homing rate was assumed as the rate of resistance allele formation. Diagram F shows a green area, representing scenarios where a subpopulation of homozygous resistant organisms would make up more than 95% of the population. In either case, the resistance begins in the low fitness/high release percentage area and spreads into the medium fitness/low release percentage area, over 40 generations.

For a quantitative comparison of the gene drives investigated using the deterministic model, areas (i.e., the number of data points, see Supplemental Information) of wild type population percentage below 5% (red) and above 95% (blue) of a respective generation were divided by each other (red/blue). The resulting curves for the ratios over 60 generations post release are shown in Fig. 5A. These curves show an asymptotic behaviour, with the exception of the CRISPR/Cas-mediated gene drive with resistance. In this special case, the number of data points of the green (resistance) areas are taken into account by subtraction from red areas. Omitting resistance allele formation results in an asymptotic behaviour similar to the other GD systems. The thresholds approached by the asymptotes can serve as a measure of the invasiveness of a technology. For a ranking of the drives investigated here, all thresholds are normalized to the highest threshold (CRISPR/Cas-based drive with no resistance formation) and listed in Table 1. Furthermore, in Fig. 5B, the fitted curves for fitness loss and fitness gain transgenes are shown as positive and negative controls on a logarithmic axis. Note, the by far higher invasiveness of a transgene conferring a fitness gain.

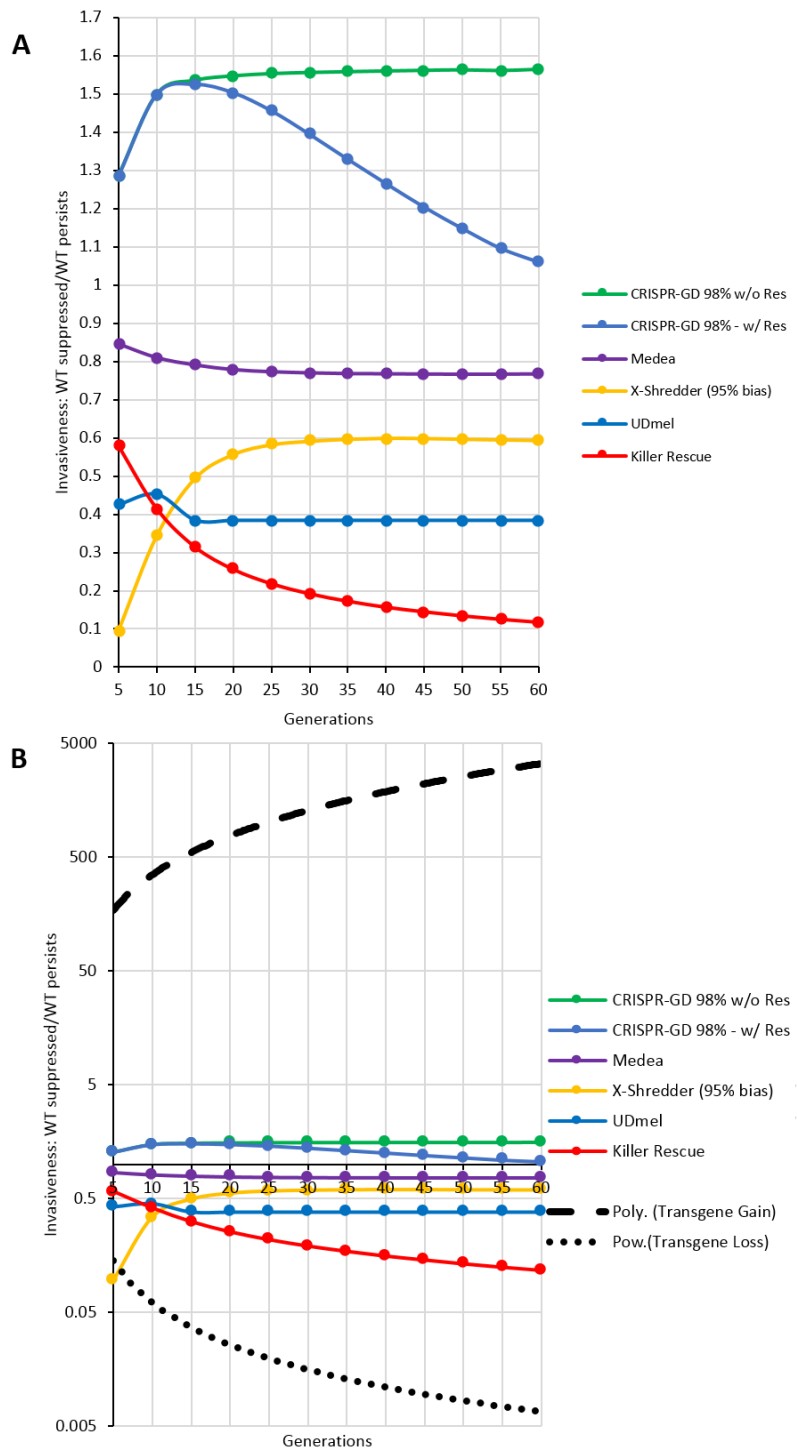

**Figure 5 Invasiveness as ratio of complete population replacement vs. loss of gene drive construct per generation.** Each gene drive technique shows an asymptotic behaviour with the exception of the CRISPR/Cas-mediated gene drives, due to the formation of resistance alleles (labelled as w/Res). A comparable CRISPR/Cas-mediated gene drive without resistance allele formation (labelled as w/o Res) shows asymptotic behaviour as well. (A) GD techniques on a decimal scale. (B) GD Techniques on a logarithmic scale compared to the fitted curves for transgenes with fitness loss (power) and fitness gain (polynomial).

**Table 1  Ranking of gene drive invasiveness.**

| Gene drive technology | Asymptote threshold | Invasiveness |
|---|---|---|
| CRISPR/Cas (without resistance formation) | 1.5644 | 100% |
| Medea | 0.7682 | 49,1% |
| Y-linked X-Shredder (95% sex bias) | 0.5941 | 29,7% |
| Two-Locus Underdominance | 0.4354 | 27,8% |
| Killer-Rescue | 0.1118 | 7,1% |

## DISCUSSION

A prospective and comparative characterization of gene drives depends on knowledge about the mode of action and corresponding technical characteristics with regard to their functional effectiveness, their potential to spread and expectable detrimental effects. As a prerequisite for further orientation on the impact (power and range) and potential ramifications of gene drives, common features of these technological approaches were selected for a comparative technology characterization as well as an analysis of factors (traits) which influence its impact, spread and invasiveness. In Table 2 the here discussed GD techniques are compared, summarizing this prospective assessment.

The comparative technology characterization revealed differences which presumably lead to different levels of expectable exposure. For instance, gene drives may employ different mechanisms to ensure super-mendelian inheritance, reaching from more or less passive drives, acting by selective embryonic lethality (*Akbari et al., 2015*) as Medea, Underdominance and Killer-Rescue or the biased segregation of sex chromosomes during meiosis (X-Shredder) to the high potential with regard to power and especially range that could be achieved with endonuclease-based gene drives using the CRISPR/Cas9-system. The release of GDOs carrying CRISPR/Cas-based drives is hard to assess due to their outstanding versatility and potential dynamics. It strongly depends on the chosen setup. Furthermore, the lack of knowledge about their potential behaviour is additionally increased by the fact that there is no naturally occurring comparative mechanism for this type of drive. Moreover, the list of potential sources of failure is comparably long for CRISPR/Cas9-based drives. With regard to exposure in particular actively replicating drives based on homing endonuclease genes (e.g., CRISPR/Cas9) are potentially highly effective in replacing a wild type population. This is reflected by the low initial intensity of intervention for CRISPR/Cas drives and its expected high range and invasiveness. The latter exceeds the results of all other drive types in our analysis. Of the passive drives, the Medea-approach showed the highest invasiveness. The lowest invasiveness was found for Killer-Rescue drives. This result corresponds with the functionalities reported in the literature that lead to a classification as potential low range-drive in the technology characterization (*Gould et al., 2008*). Apart from this rough classification it is essential to investigate the average impact of resistance formation or detrimental mutations of these drives in experimental approaches over several generations in the same target species. Their performance and hence also a ranking with regard to exposure relevant functionalities may yield a different order. In the present work, resistance formation is only scarcely characterized due to

**Table 2 Comparison of gene drive-specific characteristics.**

| | | X-Shredder | Killer-Rescue | Medea | Two-locus UD | CRISPR/Cas |
|---|---|---|---|---|---|---|
| | Mode of action | Sex ratio/ Chromosomal disruption | Toxin/Antidote | Toxin/Antidote | Toxin/Antidote | Heterozygote to homozygote conversion |
| Depth of intervention | Class | Suppression | Replacement | Replacement | Replacement | Replacement |
| | Range | High | Low | High | Low | High |
| | Invasiveness | Moderate | Low | High | Moderate | Very high |
| Intensity of intervention | | High | High | High | High | Low |
| Reliability | Resistance formation | Very unlikely | Possible | Possible | Unlikely | Likely |
| | Linkage Loss | Unknown | Unknown | Unlikely | Unlikely | Likely |
| Risk management | Reversibility | None, maybe CRISPR/Cas-based overwriting drive | Wild type release | 2nd generation Medea drive | Wild type release | Overwriting drive |

preliminary information from first experiments or derived from the design of the drive. However, in particular CRISPR/Cas-drives tend to be afflicted with a comparably large number of sources for potential mutations of either their own genetic information or its target site. This instability is also the reason for a potential linkage loss of CRISPR/Cas drives with their cargo. However, a recent work by *Kyrou et al. (2018)* shows that resistance formation due to non-homologous end joining repair can be effectively circumvented by targeting highly conserved essential genes.

The more power and range are increased, the greater becomes the extent of uncertainties and ignorance with (a) the magnitude of known unknowns regarding potential effects of known dependencies and relationships of the target species and possibly affected non-target species and (b) not yet determinable effects (unknown unknowns) due to extreme exposure (spread and invasion). The latter may arise from as yet unknown interactions and the inherent instability of genetic information which becomes more relevant with increasing numbers of GDOs and ongoing evolution. Although meanwhile a number of options for limitation of spread and invasion or reversibility of gene drives are brought into the discussion (cp. Table 2) including an initial laboratory-scale trial of a split drive in yeast (*DiCarlo et al., 2015b*), they still represent theoretical options. A proof of principle in a relevant scale is still lacking and even for the split drive a limitation of its spread in comparison with the equivalent unseparated drive was not yet shown.

## CONCLUSIONS

Gene drives constitute a tipping point in the technological development of genetic engineering, due to their inherent capability to spread and invade to either suppress or replace a naturally occurring population. This new functionality separates GDOs from other GMOs released into agricultural ecosystems, so far. Due to the by trend high spatio-temporal range of a released GD, exposure and thereby non-knowledge about possible consequences increases, reaching from enormous scientific uncertainties to vast ignorance.

A GD is a technology capable to reproduce itself and potentially undergo mutational changes over time. Not only do GDs affect the environment, the environment affects the GDs as well. Thus, a GD engineered in the laboratory, once released will be confronted with evolutionary processes. It is hard to predict how the genetic information of the drive will be influenced by mutation and selection processes post-release. The ecological and evolutionary level of complexity presents a complicated interwoven web of biotic and abiotic factors. The initially small possibility for such effects rises enormously with the depth and intensity of interventions into these systems. A correspondingly extended risk assessment (hazard and exposure assessment) is required to fathom the extent and kind of non-knowledge on the different organizational levels of biosystems that are likely to be affected by GDs. It is questionable whether our criteria, methods and models for prospective and comparative assessment are already sufficient to adequately investigate exposure potentials of released GDs. Moreover, for the prospective and comparative assessment of hazards there still exists a lack of adequate criteria, methods and models.

Besides the spread of new functionalities, the effects of a strong reduction of populations up to their eradication may evoke complex ecological changes that have to be considered. With regard to risk management, there are no proven management options to mitigate or reverse a once released GD. And even if we look at some proposals, just as the drive needs several generations to establish itself, so will the rescue drive. With concrete plans for the release of GDOs, important questions regarding technological, ecological as well as ethical issues become apparent. There may be great potential benefits but there are also very high reasons for concern.

In this work, we presented an overview of different GD techniques which are planned to tackle infectious diseases, invasive species or pest organisms. In order to develop an approach for prospective risk assessment we examined GD techniques, comparatively characterizing their depth (above all range) and intensity of intervention, reliability and possible corrective actions. Furthermore, based on their inheritance schemes we presented an assessment of the drive's efficiencies to invade a population that illustrates the high capacity of CRISPR/Cas9-based drives. The analysis revealed a differentiated picture with regard to the potential range of the investigated drives. However, as long as this characterization has to rely on theoretical assumptions or initial experimental results in the laboratory scale, the present assessment should not be used as a basis to choose a "safer" GD for release. It rather delivers an early description of the drives specificities that may be a useful contribution for efforts to outline low-risk development paths. There is an urgent demand for further prospective analysis of impact, side effects, countermeasures and the feasibility of low-risk approaches. In some respects, existing risk assessment for pathogenic viruses resembles the criteria of technology characterization. For example the Influenza Risk Assessment Tool (IRAT) (*Cox, Trock & Burke, 2014*) of the US-Centers for Disease Control and Prevention (CDC) at least partially apply related criteria for exposure assessment in the categories 'properties of the virus' and 'ecology and epidemiology' (also see on the homepage of the CDC: https://www.cdc.gov/flu/pandemic-resources/national-strategy/risk-assessment.htm; last accessed 2019 Feb 06). In the light of the extensive experience with viral infections it should be worth a try to consider the significance of the criteria for viral spread for a further development of the criteria applied in technology assessment of GDs.

## ACKNOWLEDGEMENTS

The authors thank Bartholomäus Gruchalski and Dr. Rolf Roth for their help with the mathematical approach for invasiveness.

### Funding

This work has been funded by the German Ministry for Science and Research (BMBF) within the study "Bio Tip pilot study: Genetic innovations as triggers of phase transitions in population dynamics of animals and plants (Gene Tip)" under the code 01LC1724A.

The funders had no role in study design, data collection and analysis, decision to publish, or preparation of the manuscript.

### Grant Disclosures

The following grant information was disclosed by the authors:
German Ministry for Science and Research (BMBF).

### Competing Interests

The authors declare there are no competing interests.

### Author Contributions

- Johannes L. Frieß conceived and designed the modelling and prepared the application of of prospective and precautionary technology characterization for gene drives, performed the modelling, analyzed the data, contributed materials/analysis tools, prepared figures and/or tables, authored or reviewed drafts of the paper, approved the final draft.
- Arnim von Gleich conceived and designed the methodological approach of prospective and precautionary technology characterization authored or reviewed drafts of the paper, approved the final draft.
- Bernd Giese co-developed the methodological approach of prospective and precautionary technology characterization and its application for gene drives, contributed prepared figures and/or tables, authored or reviewed drafts of the paper, approved the final draft.

### Data Availability

The Supplemental File contains all calculations necessary for the computation and analysis of the model presented in the article.

### Supplemental Information

Supplemental information for this article can be found online at http://dx.doi.org/10.7717/peerj.6793#supplemental-information.

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
