# Peer review of "Gene drives as a new quality in GMO releases—a comparative technology characterization"

_PeerJ, doi:10.7717/peerj.6793_

## Round 0.1 · original submission · Minor Revisions

Dear author

Your paper has been assessed by two reviewers and myself as academic Editor.

As you could see below, the manuscript was rated very positively by all reviewers, with only minor revisions suggested. I congratulate you for the nice piece of work, which will add value to PeerJ.

I personally found the work very useful and well written. The results will certainly be appreciated by the community working on GMO Biosafety/Risk assessment.

My personal suggestion to improve the manuscript is to add some sentences regarding relevant genetic differences among plant and animal GMOs (e.g. plants spread pollen and are mostly bisexual individuals). Most releases of genetically modified organisms (GMOs) are primarily plants, and remarkably, the biosafety concerns of the public have focused more on plants than on transgenic animals or inclusive transgenic bacteria. The word "plants" is only mentioned in the abstract, but does not appear anymore in the rest of the manuscript, but instead Drosophila, Mus, Culex, etc.

I would also suggest to make an analogy between gene drives (invasive selfish genes), with that of infectious virus. Maybe it is helpful to employ risk assessment strategies developed against virus pandemics' (e.g. Ebola, HIV, influenza) and consider them for worldwide prevention (paradigm shift in the handling of GMOs).

Reviewer 1 ·

Basic reporting

The authors use unambiguous and professional English in their paper. In some cases technical/ juristic terms should be used or defined more clearly. E.g. in line 24 it is stated the experimental field test would be tantamount to the first deliberate release. However the juristic term “deliberate release” applies per definition to any experimental release at least in European legislation. In line 120 the “precautionary principle” is introduced however it is not referenced or clearly defined what is meant by it (general ethical rule or legally based). Since the “precautionary principle” is used as an important point in later argumentation, this should be more elaborated. It would also help understanding, if central terms like “depth of intervention” clearly defined and would be referenced also by third party literature.

The context and background of the underlying technologies and relevant problems are very good introduced and placed into context. Literature is generally well referenced and relevant. However at some points referencing could be broadened. E.g. in line 105 only literature co/authored by the authors is cited third party literature might strengthen creditability. In line 423 a more general picture of homing efficiencies could be drawn by citing broader literature (e.g. Ganz and Bier2015, KaramiNejadRanjbar 2017).
Literatur on other available models to analyse gene drives should be provided.

Structure conforms good to PeerJ standards. At some point it is not clear if text and statements are placed in the right topic. Line 123/124 seems to state a rather a result of a technology characterization than a method or a part thereof. Line 120 to 123 seem to rather state an opinion, or conclusion than a method. If named focus of technology characterization is factual part of the method it should be referenced by literature. In my opinion a scientific characterization should be open ended and does not include conclusions and actions that may be based on its results (“lies on the prevention of..”).
Important parts of the results

In Fig. 3 and Fig. 4 shading oft he colors is not explained despite being seemingly the same graphs, on is named „cross sections“ the other „ cross section overlays“ .
Table 1 lists “Pervasivness”, a term that is not referenced anywere in the the rest paper or other literature.
Fig. 5 it would be reasonable and helpful to include here results from the positive and negative controls displayed in Fig. 3.

Raw data of presented model are provided.

Experimental design

The paper is mostly a literature review supplemented by a basic deterministic inheritance model that visualises gene drive inheritance.
The research question has been well defined and is relevant and the goal to fill a knowledge gap clearly identifiable.
The methods are described with sufficient detail, however central information with respect to the model is to be found in supplemental material.

Validity of the findings

The paper mostly aggregates findings from other papers.
Were the model is concerned the authors might expand on the reasoning for chosing this type of model of other models available in literature.

As positive and negative standards have been introduced to test the model, this results should also be included and put into relation to technologies characterised by the model in the section Discussion and conclusion. E.g. in Fig 5

·

Basic reporting

Very minor comments presented to the authors in the section for General Comments

Experimental design

No problems found

Validity of the findings

Very minor comments presented to the authors in the section for General Comments

Additional comments

It is my opinion that, the manuscript presented by Frie et. al. is of sufficient quality and significance that should be published. It presents a very good assessment of the properties of different Gene Drive systems that should be taken into consideration when performing a risk assessment of GM organisms based on any of these systems.

Minor comments.

1. Line 17: Compared to previous releases of genetically modified organisms (GMOs) which were primarily plants, gene drives represent a paradigm shift in the handling of GMOs: Current regulation of the release of GMOs assumes that for specific periods of time a certain number of GMOs will be released in a particular region.

This statement does not recognize that, in some instances, the release of a current GMO could lead to the transformation of entire wild populations. If a GMO plant is released in its center of origin/diversity, and there are plenty of sexually compatible relatives, the transgene in principle can spread into the entire wild population -of course depending on the final phenotypes, selective advantage/pressure and other factors. This is important to consider because -albeit with significant differences to the GD issue- the research community working on risk assessment and regulators have already been working with this problem, and methodology including models to simulate the introgression of transgenes into wild populations have also been used.

I believe this issue merits comments and recognition by the authors.

2. Although it is acknowledged that the use of gene drives could lead to the disappearance of entire populations, that is not always the intention. In Line 64 we read “In agriculture, weeds and crop pests could be eradicated” and indeed this may be the case, however, the prevalent idea is to use GD to revert populations of weeds and insects that have acquired resistance to efficient and safe herbicides and pesticides, to the sensitive phenotype, to again allow the use of those herbicides and pesticides. So at least such applications should also be mentioned even if these are not considered and discussed in detail because that is not the intention of the research reported.

3. Comment on line 645: In the light of the absence of proven options to a) correct
potential damage or b) just to limit the inherently self-propagating mechanism of gene drives, these functionalities reveal important ‘reasons for very high concern’ with regard to the requirements of the precautionary principle.

I would like to recall what is mentioned in Annex III of the Cartagena Protocol -derived from the Convention of Biodiversity and aligned to the Precautionary principle:

General principles
4. Lack of scientific knowledge or scientific consensus should not necessarily be interpreted as indicating a particular level of risk, an absence of risk, or an acceptable risk.

6. Risk assessment should be carried out on a case-by-case basis. The required information may vary in nature and level of detail from case to case, depending on the living modified organism concerned, its intended use and the likely potential receiving environment.

Methodology
7 (f). Where there is uncertainty regarding the level of risk, it may be addressed by requesting further information on the specific issues of concern or by implementing appropriate risk management strategies and/or monitoring the living modified organism in the receiving environment.

Therefore, I believe that such broad and general statements are not acceptable. I strongly believe that the community of risk assessors/regulators who will read this paper will be aware of the risks associated with the technology -clearly presented by the authors- and will be taken into consideration on any environmental risk assessment of GDO, among many other considerations required. This is in line with the more balanced assessment presented by the authors in their closing paragraph:

“However, as long as this characterization has to rely on theoretical assumptions or initial experimental results in the laboratory scale, the present assessment should
not be used as a basis to choose a “safer” gene drive for release. It rather delivers an early description of the drives specificities that may be a useful contribution for efforts to outline low-risk development paths. There is an urgent demand for further prospective analysis of impact, side effects, countermeasures and the feasibility of low-risk approaches.”

---

## Round 0.2 · accepted · Accept

Dear authors

I can read that you have addressed all the minor corrections.
The Manuscript has been impoved.

I congratulate you for the nice piece of work, which will add value to PeerJ.

#